# Systematic Review on Saliva Biomarkers in Patients Diagnosed with Morbus Alzheimer and Morbus Parkinson

**DOI:** 10.3390/biomedicines10071702

**Published:** 2022-07-14

**Authors:** Michael Wolgin, Magdalena Zobernig, Valentyn Dvornyk, Ralf J. Braun, Andrej M. Kielbassa

**Affiliations:** 1Center for Operative Dentistry, Periodontology, and Endodontology, Department of Dentistry, Faculty of Medicine and Dentistry, Danube Private University, 3500 Krems an der Donau, Austria; michael.wolgin@dp-uni.ac.at (M.W.); zobernig.magdalena@dp-uni.eu (M.Z.); andrej.kielbassa@dp-uni.ac.at (A.M.K.); 2Department of Prosthetic Dentistry and Implantology, Faculty of Dentistry, Poltava State Medical University, 36011 Poltava, Ukraine; international@pdmu.edu.ua; 3Research Division for Neurodegenerative Diseases, Center for Biosciences, Department of Medicine, Faculty of Medicine and Dentistry, Danube Private University, 3500 Krems an der Donau, Austria

**Keywords:** Alzheimer’s disease, amyloid-β, α-synuclein, biomarker, DJ-1, Parkinson’s disease, saliva, tau

## Abstract

Extracellular plaques composed of the hydrophobic peptide amyloid-β and intraneuronal accumulation of the hyperphosphorylated protein tau (p-tau) are pathological hallmarks found in the brains of most people affected by Alzheimer’s disease (AD). In Parkinson’s disease (PD), Lewy bodies, i.e., intraneuronal protein deposits comprising the protein α-synuclein, are a typical disease feature. As these hallmarks located in the brain are hardly traceable, reliable biomarkers from easily accessible body fluids are key for accurate diagnosis. The aim of the present work was to review the available literature regarding potential biomarkers of AD and PD in the saliva. The databases PubMed, Google Scholar, LILACS, LIVIVO, VHL regional portal, Cochrane Library, eLIBRARY, and IOS Press were consulted for the literature search. Screening of titles and abstracts followed the PRISMA guidelines, while data extraction and the assessment of full texts were carried out in accordance with the Newcastle–Ottawa Scale assessment. The review shows significant increases in levels of the amyloid-β Aβ1-42 and elevated p-tau to total tau (t-tau) ratios in salivary samples of AD patients, in comparison with healthy controls. In PD patients, levels of α-synuclein in salivary samples significantly decreased compared to healthy controls, whereas oligomeric α-synuclein and the ratio of oligomeric α-synuclein to total α-synuclein markedly increased. Salivary biomarkers represent a promising diagnostic tool for neurodegenerative diseases. Further high-quality case–control studies are needed to substantiate their accuracy.

## 1. Introduction

Alzheimer’s disease (AD) is the most common neurodegenerative disease of the central nervous system, manifesting as dementia, confusion, and cognitive impairment. In such cases, markers, such as amyloid-β (Aβ) peptides and tau proteins, seem to be responsible for the loss of neurons at the hippocampus, basal forebrain, and other cortical areas [1,2]. Aβ peptides derive from enzyme proteolysis of the amyloid precursor protein (APP), and play physiological roles in memory formation and lipid homeostasis, regulation of neuronal activity, and neurite growth [3]. AD is characterized by an excessive accumulation of amyloid-β and neurofibrillary tangles, leading to neuronal damage due to interrelated pathological processes [1,2]. Pathologically, APP is cleaved by β- or γ-secretase, which produces a fragment of 99 amino acids (AA) and cleaves the fragment into one peptide with 40 AA (Aβ1-40), and one with 42 AA (Aβ1-42). In the extracellular space, the peptides assemble into insoluble oligomers and fibrils, which form an amyloid plaque, triggering an inflammatory process that leads to neuronal damage [1,2]. Tau proteins are microtubule-associated proteins responsible for axonal transport and neuronal structure, as well as plasticity, leading to the stabilization of microtubules. The function of tau is regulated by the balance between de- and phosphorylation, which can be altered by mutations in the tau protein sequence, affecting the phosphorylation site of the protein, inducing hyperphosphorylation. The hyperphosphorylated tau (p-tau) can accumulate and form intracellular neurofibrillary tangles, which then synergize with Aβ, increasing their cytotoxic functions [1,2]. In diseases such as Creutzfeldt–Jakob disease (CJD) or frontotemporal dementia, high levels of total tau (t-tau) are also found in cerebrospinal fluid (CSF) samples. Consequently, t-tau cannot be designated as a determinant biomarker for AD [4]. Phosphorylated tau (p-tau), on the other hand, could be used to distinguish AD from other types of dementia, because it shows the phosphorylation of tau, and thus, the possible shape of bundles in the brains of AD patients. So far, at least 30 phosphorylation sites on the tau protein are known; the most common are threonine 181, threonine 231, serine 199, serine 396, and serine 404 [5].

Parkinson’s disease (PD) is the second most common neurodegenerative disorder, which results in multiple motoric and cognitive deficits [6,7]. Approximately 90% of PD patients suffer from an idiopathic form, with the remaining 10% having a familial background [8]. PD of familial origin is characterized by autosomal dominant and recessive mutations in different genes. Among other symptoms, PD is clinically characterized by tremors, muscle rigidity, postural instability, akinesia, bradykinesia, anxiety, depression, sleep disturbance, dementia, and psychosis, which result from a pathophysiological loss of dopaminergic neurons from the substantia nigra, and concomitant reduction in dopamine production in the corpus striatum. These pathophysiological processes involve α-synuclein, which regulates, among other things, dopamine release. This protein is one of the main components of Lewy bodies, the characteristic structures found in the brain tissue of PD patients [6,8]. PD pathology is, therefore, attributed to increased levels of α-synuclein in neurons due to overexpression of the α-synuclein-encoding gene *SNCA*, and increased gene copy number, together with different point mutations, affecting α-synuclein and resulting in oligomeric or fibrillar aggregations. In contrast to the physiologically dominating monomeric form of α-synuclein, its oligomeric form (converted into amyloid fibrils to form Lewy bodies) predominates in patients suffering from PD [6,8]. DJ-1, a small, highly conserved protein consisting of 189 amino acids, is another PD-associated biomarker. Being ubiquitously expressed, DJ-1 exists in dimeric form under physiological circumstances. Its mutation at the *PARK7* gene is associated with early onset, familial, autosomal recessive PD [9]. DJ-1 is localized in the cytoplasm and, to a lesser extent, in the mitochondria, as well as in the nuclei of dopaminergic neurons. The monomers dimerize under oxidative stress and translocate to the mitochondria, after which they migrate to the nucleus [10]. DJ-1 serves as a molecular chaperone to inhibit the formation of α-synuclein fibrils. This function represents an essential step in the formation of α-synuclein oligomers, which play a key role in PD pathology [11].

Changes in brain function and structure such as atrophy, structural changes, plaques, inflammation, and oxidative stress are known signs for clinical diagnosis, and can be evaluated by imaging techniques such as computed tomography (CT), magnetic resonance imaging (MRI), functional magnetic resonance imaging (fMRI), or positron emission tomography (PET) using contrast agents [12,13,14,15]. However, cerebrospinal fluid (CSF) analysis has proved to be a more direct and effective diagnostic method. Being directly associated with central and peripheral nervous tissue, CSF contains specific markers that have an impact on the diagnosis of neurodegenerative diseases; therefore, the CSF is currently the most used diagnostic tool for recognizing AD and PD. However, sampling is associated with complex methods, such as lumbar puncture—a specialized, invasive, and relatively harmful procedure, accompanied by pain and side effects. In addition, the patient is subjected to stress during the procedure, which can be associated with increases in cortisol, affecting the measurement of biochemical parameters [12,13,14,15]. A less invasive method is the analysis of serum and plasma, with advantages such as low risk and ease of obtaining blood samples. However, the effective use of plasma is limited because of difficulties in transportation, standardization, and reproducibility of results due to the fluctuating proteome of blood. Due to these limitations, more non-invasive methods are being sought. Saliva can represent another source of readily available biomarkers [12,13,14,15].

Saliva samples have proved to be easy to obtain, inexpensive, non-invasive, and compatible with various analytical assays. Thus, saliva sampling could be a legitimate alternative to other diagnostic methods, such as CSF; moreover, valid and reproducible salivary biomarkers would be preferable to those obtained via CSF or plasma for the aforementioned reasons [12,13,14,15]. Therefore, the aim of the present systematic review was to summarize available studies on salivary biomarkers for the diagnosis of neurodegenerative diseases, such as AD and PD, and to investigate whether differences regarding the concentrations of salivary biomarkers, amyloid-β and tau for AD and α-synuclein and DJ-1 for PD, can be observed in patients suffering from these neurodegenerative diseases compared to healthy individuals.

## 2. Materials and Methods

For the identification of studies to be considered for this review, PubMed/Medline, Google Scholar, LILACS, LIVIVO, VHL regional portal, Cochrane Library, eLIBRARY, and IOS Press databases were searched from 10 January 2022 to 30 April 2022. The electronic search in PubMed/Medline was adjusted in terms of applying the most relevant search terms in combination with an adequate Boolean search algorithm: ((Alzheimer’s disease) AND (Amyloid Beta)) OR (Tau Protein) AND (Saliva), ((Parkinson’s Disease (MeSH Terms)) AND (Saliva (MeSH Terms))) AND (α-synuclein (MeSH Terms)), with filters from 2010–2022. Similar search strategies were developed for further databases used in the present systematic review. To cover potentially eligible literature sources without English-language abstracts (e.g., publications from CIS countries, or former USSR) and to perform the corresponding electronic literature search in eLIBRARY, the used MeSH terms were translated into Russian. All MeSH terms were finalized by mutual agreement between the first (MW) and the second author (MZ) of the present review. Moreover, to revise for possible additional papers in all available languages (e.g., in German, Russian, or Chinese), the reference lists of identified and relevant studies on the subject were reviewed. This combination of information sources retrieved both published journal articles and grey literature (e.g., dissertations, diploma theses, study register entries). To detect studies not found in databases, reference lists of included studies and applicable reviews were examined, and citations were manually searched on the websites of relevant journals. The search was conducted in accordance with PRISMA guidelines for systematic reviews [16], and without language restriction. However, only publications in English and Chinese were found. The protocol (Registration number CRD42022307546) of the present review was pre-registered at the Prospero Register of Systematic Reviews (PROSPERO) on the 3rd of March 2022. The authors framed an answerable and researchable study question to the established PICO (T) format (Population/Patient/Problem, Intervention, Comparison, Outcome): “For patients suffering from AD and PD (Problem/Patient), will an effect of clinically diagnosed disease (Intervention) as compared to an absence of disease (Control/Comparison) result in a comparable salivary concentration of biomarkers amyloid-β, tau, α-synuclein, and DJ-1 (Outcome)?”.

Each of the abstracts found was evaluated, and then only original publications related to biochemical properties of human saliva and saliva sampling were considered. To deal with possible heterogeneity of results, only case–control studies assessing concentrations of amyloid-β, tau, α-synuclein, and DJ-1 by means of quantitative assays, such as ELISA, Western blot, and similar methods, were included in the present review, while investigations on other salivary biomarkers or investigations with non-quantitative assays were considered as irrelevant and were, therefore, excluded. In addition, reviews, meta-analyses, studies that examined the microbiome and oral hygiene, mechanisms of dysphagia, treatment of hypo- and hypersalivation, and other clinical interventions that focused exclusively on unique changes in salivary composition associated with AD and PD, as well as studies that examined non-specific aspects of salivary composition, such as electrolyte content or total protein concentration, were also excluded from the further analysis. The selection process is illustrated in a PRISMA flow diagram (Figure 1). Two independent reviewers (MW and MZ) screened the titles and abstracts of all the identified studies, immediately excluding obviously irrelevant works. Then, both reviewers independently reviewed the remaining full-text articles, and selected relevant publications (Table 1 and Table 2 for AD; Table 3 and Table 4 for PD) based on the abovementioned inclusion and exclusion criteria, followed by independent extraction of data from each eligible study. Subsequent data synthesis and analysis by means of Review Manager (RevMan, version 5.4.1.; Cochrane: London, UK, 2020) showed diversity and heterogeneity of data in the included studies. Therefore, sensitivity analyses were not feasible, and only a narrative synthesis of the presented data was possible.

For articles selected in the present review, their quality was assessed using the modified Newcastle–Ottawa Scale (NOS) [17]. According to this, the assessment was performed in three different areas: selection of study groups; comparability of groups and determination of exposure or outcome depending on the study type; quality of outcome and adequacy of follow-up, with a maximum score of 9 points. Studies with Newcastle–Ottawa Scale scores of 0 to 3, 4 to 6, and 7 to 9 were classified as having high, moderate, and low risk of bias, respectively.

## 3. Results

### 3.1. Studies on Salivary Biomarkers Amyloid-β and Tau in AD Patients

The initial electronic search resulted in 453 references. Additional records (*n* = 8) were identified through other sources, such as national and international dissertation databases or by the establishment of personal communication with authors and working groups. All duplicates were first excluded. In the end, 25 studies remained after screening for possible bias of the abstracts and titles. After reading the full-text versions and adhering to predetermined inclusion requirements, ten studies remained, while fifteen studies were excluded from the subsequent analysis. The most common reasons for the exclusion were: studies for meta-analyses used inconsistent measurement methods and units of measurement, and in some cases, there was missing information on the sex of the groups studied, and limited outcome data. The study selection process is summarized in Figure 1a.

#### 3.1.1. Quality Assessment of Included Studies for Biomarkers Amyloid-β and Tau

In accordance with the Cochrane Reviewers’ Handbook, the studies were assessed and graded to limit the risk of bias caused by inadequacies in study design, conduct, or analysis. In this case, each study was rated on three different levels according to the Newcastle–Ottawa Scale (NOS). Of the nine included studies concerning the salivary concentration of AD-relevant biomarkers, only one received the score “good quality”, while five investigations were of “fair quality”, and one study was considered of “low quality”. The results of the NOS scoring assessment are given in Table 5. Reasons for categorizing the studies as moderate quality were mostly due to the lack of sample size calculations, limitations in design (i.e., non-observance of confounding factors and lacking sound representativeness), or inaccuracy in elucidating the obtained data, such as disregarded non-respondents data.

#### 3.1.2. Compilation, Characteristics, and Outcome of Included Studies for Biomarker Amyloid-β

Eight out of ten studies on saliva biomarkers for AD aimed to determine the levels of Aβ42 in this body fluid [18,19,20,21,22,23,24,27] (Table 2). As two of the studies were unable to detect this hydrophobic peptide [19,21], six studies remained for further analysis [18,20,22,23,24,27].

The first study, published in 2010, quantified salivary and plasma concentrations of Aβ40 and Aβ42 in 70 AD patients and 56 study participants in the control group (healthy subjects) by means of enzyme-linked immunosorbent assay (ELISA) (Biosource International, Invitrogen, Carlsbad, CA, USA) [18]. The authors demonstrated significantly elevated Aβ42 levels in the saliva samples of AD patients (6.81 ± 20.04 pg/mL) compared to the corresponding control group (2.89 ± 4.96 pg/mL). In contrast, data concerning Aβ40 levels in saliva showed no significant differences between samples of AD patients and healthy subjects. Consequently, the ratio of Aβ42/Aβ40 was found to be elevated in the saliva, when comparing AD patients to the control group, although this increase was not statistically significant. Notably, neither plasma levels of Aβ40 nor Aβ42 showed any statistically significant differences among these groups. Additionally, the authors investigated the relationship between the salivary concentration of Aβ42 depending on severity levels of AD. In this instance, when compared with healthy subjects, a significant elevation in Aβ42 in patients with mild (7.67 ± 16.25 pg/mL) and moderate (11.70 ± 34.76 pg/mL) forms of AD could be detected. Interestingly enough, Aβ42 levels corresponding with severe AD (3.03 ± 3.49 pg/mL) demonstrated considerably lower values than the other two groups, mentioned above, being comparable with physiological concentrations of Aβ42. The authors particularly emphasize the importance of acknowledging the fact that the determination of Aβ42 in CSF of AD patients, although included in this study, was unreliable.

Kim et al. investigated, in 2014, the salivary concentrations of Aβ42 and Aβ40 using a magnetic immunoassay MIA (Chemicell, Berlin, Germany) with 45 study participants, including 17 healthy individuals and 28 AD patients with mild and severe AD forms [20]. This investigation demonstrated that there are significantly elevated Aβ42 concentrations (approximately 2000 pg/mL) in the saliva of patients with severe AD in comparison with moderate sufferers and healthy individuals. In contrast, the observed increase in salivary levels of Aβ40 turned out to be non-significant. Estimated using the data presented by these authors, the ratio of Aβ42/Aβ40 should be elevated when comparing patients suffering from severe AD with patients affected by the mild form. Furthermore, to verify the results of MIA, an additional experiment using conventional ELISA (Covance, Dedham, MA, USA) was conducted, which demonstrated a similar trend (data not shown in the original study).

Lee et al. quantified, in seven AD and three pre-AD patients, and in addition to the salivary concentration of Aβ42, levels of this amyloid in samples of different organs, such as the kidney, small intestine, pancreas, spleen, hippocampus, and sensory cortex by means of customized ELISA kits (Biosource International, Invitrogen, Carlsbad, CA, USA) [22]. In this case, the results were compared with corresponding Aβ42 concentrations in 27 healthy subjects, serving as a control group, however, one of them was a PD patient. The organs of AD patients, in particular the spleen, pancreas, and kidney, showed significant increases in Aβ42 level compared to the control group. Notably, salivary concentrations of Aβ42 in AD patients (59.07 ± 6.33 pg/mL) compared to the control group (22.06 ± 0.41 pg/mL) revealed a similar and significant pattern. Although Aβ42 in saliva exhibited lower concentrations compared to the organs mentioned above, these authors demonstrated that ELISA might be considered as a suitable method to detect even lower Aβ42 levels in the saliva.

A follow-up study from the same group summarized the results obtained from the previous investigation in the form of a review, expanding the study with new data from an additional experiment with 31 control group participants and 23 AD patients in total [23]. As before, the Aβ42 concentrations measured in saliva of AD patients (53.95 ± 2.24 pg/mL) were significantly increased compared to the samples from healthy subjects (26.55 ± 1.85 pg/mL) (mean values from healthy subjects were calculated and statistical analysis was performed with the original data provided in the paper). In addition, the authors divided the control group, consisting of healthy subjects, into so called “low controls” with low Aβ42 concentration (21.54 ± 0.19 pg/mL) and “high controls” with enhanced Aβ42 levels in saliva (45.96 ± 3.01 pg/mL). Since the average Aβ42 concentration measured in saliva of AD patients (53.95 ± 2.24 pg/mL) was comparable with the corresponding concentration in “high controls”, the authors assumed the identification of persons at risk of AD within this control group.

In 2018, another ELISA-based (Aurin Biotech, Inc., Vancouver, BC, Canada) study using in 22 participants (15 with AD, and 7 in the control group) reported comparable results concerning significantly elevated concentrations of Aβ42 in saliva of AD sufferers (51.7 ± 1.6 pg/mL) vs. the control group (21.1 ± 0.3 pg/mL) [24].

Finally, in 2020, authors further quantified salivary concentrations of Aβ40 and Aβ42 in 69 AD patients and 83 healthy subjects by means of MILLIPLEX^®^ MAP (Human Amyloid Beta and Tau Magnetic Bead Panel—Multiplex Assay; Life Science, Darmstadt, Germany) [27]. In contrast with the previous studies, the Aβ42 concentrations measured in saliva of AD patients (3.15 ± 0.72 pg/mL) were slightly decreased compared to the samples from healthy subjects (3.57 ± 0.93 pg/mL). Data concerning Aβ40 levels in saliva showed a non-significant elevation in AD patients (21.98 ± 16.94 pg/mL) compared to the control group (19.97 ± 6.35 pg/mL).

To sum up, in five out of six studies, in which Aβ42 could be detected in the saliva, this hydrophobic peptide was either significantly [20,22,23,24] or markedly [18] elevated in AD patients compared to healthy subjects. In only one study was Aβ42 found to be lower in the saliva of patients with AD vs. the control group [27]. In contrast, Aβ40 was analyzed and detected in only three studies, with inconsistent results [22,24,29]. Thus, Aβ42 is a potential saliva AD biomarker.

#### 3.1.3. Compilation, Characteristics, and Outcome of Included Studies for Biomarker Tau

In addition to salivary Aβ levels, five out of ten studies on saliva biomarkers for AD aimed to determine the levels of hyperphosphorylated tau (p-tau) or total tau protein (t-tau) [19,21,25,26,27] (Table 2). Thus, Shi et al. published, in 2011, quantified p-tau, t-tau and Aβ42 levels in saliva from 21 AD patients and 38 control group participants using immunoprecipitation (IP), two mass spectrometers (MS) (Applied Biosystems, Foster City, CA, USA and Thermo Fisher Scientific Corp, San Jose, CA, USA), and a Luminex assay (Qiagen, Valencia, CA, USA) [19]. While Aβ42 remained undetectable in all samples, both p-tau and t-tau could be reliably detected. Whereas t-tau remained unaltered between saliva samples of AD patients compared to the control group, p-tau markedly increased, and the ratio of p-tau/t-tau significantly increased in AD patients compared to healthy subjects. Although the extraction of appropriate data was not possible, these results demonstrate that mass spectrometry and the Luminex assay might also be considered as suitable methods to detect certain AD biomarkers in saliva.

In 2015, ELISA (Biosource International, Invitrogen, Carlsbad, CA, USA) was used to investigate saliva samples from 20 AD patients and 20 healthy participants, the latter serving as a control group [21]. The results of the study show agreement with the previously described investigation: Aβ42 turned out to be undetectable in all samples, t-tau remained unaltered in AD patients compared to healthy controls, and p-tau was found to be moderately increased when comparing the AD with the control group. When the ratio of p-tau/t-tau was calculated from data extracted from the graphs of the paper, the ratio should also increase in AD patients compared to healthy subjects. However, these interpretations must be considered with caution, as the increase in p-tau was not significant, and the extraction of data from graphs is very imprecise.

Difficulty regarding sufficient data extraction was also present in this next study, published in 2019, which examined the p-tau/t-tau ratio at various phosphorylation sites (S400/T304/S404, T181, S396, S404) in the saliva of 46 AD patients, 55 patients with mild cognitive impairment (MCI), and 47 control group participants, using Western blot analysis (manufacturer details not shown) [25]. A significant increase in the p-tau/t-tau ratio in AD patients could be detected in three out of the four analyzed phosphorylation patterns (i.e., S400/T304/S404, S396, S404). Comparatively, no differences in the p-tau/t-tau ratio could be detected in CSF samples from AD patients. Consequently, this investigation shows that p-tau/t-tau ratios taken from CSF samples do not correlate with saliva samples.

One study conducted on 53 AD patients, 68 subjects with MCI, and 160 elderly healthy participants using ultrasensitive single-molecule array technology (SIMOA) (Quanterix, Lexington, MA, USA) allowed for data extraction [26]. However, the analysis was limited to t-tau only. The results show no significant differences in salivary concentration of t-tau in AD patients (12.3 ng/L) compared to the control group (9.6 ng/L).

Tvarijonaviciute et al. not only quantified Aβ42 and Aβ40 levels in saliva, but also t-tau and p-tau 181 levels, in 69 AD patients and 83 healthy subjects using MILLIPLEX^®^ MAP (Life Science, Darmstadt, Germany) [27]. As observed in the previous studies, t-tau remained unaltered between saliva samples of AD patients (21.57 ± 22.11 pg/mL) and the control group (21.15 ± 16.58 pg/mL). The p-tau levels in saliva of AD sufferers (40.33 ± 42.95 pg/mL) showed a non-significant decrease compared to the healthy subjects (42.5 ± 38.35 pg/mL).

To sum up, out of five studies investigating tau an as AD biomarker [19,21,25,26,27], salivary p-tau values were shown in four of these studies [19,21,25,27]. Of these studies, p-tau was markedly elevated in two [21,25], significantly increased in one [19], and non-significantly decreased in one [27], when comparing AD patients with healthy control subjects. In these four studies, t-tau values were shown [19,21,26,27]; t-tau consistently remained unaltered in AD patients compared to controls in all four studies. The p-tau/t-tau ratio was shown, or could be deduced from published data, in three of the studies [19,21,25]. The p-tau/t-tau ratio markedly increased in one [21], and significantly increased in two studies [19,25]. Thus, both p-tau and the p-tau/t-tau ratio are potential AD biomarkers.

### 3.2. Studies on Salivary Biomarkers α-Synuclein and DJ-1 in PD Patients

The initial electronic search resulted in 287 references. Additional records (*n* = 3) were identified through other sources, such as national and international dissertation databases or by the establishment of personal communication with authors and working groups. In the end, 20 studies remained after screening on possible bias of the abstracts and titles. After reading the full-text versions and adhering to predetermined inclusion requirements, nine studies remained, while fifteen studies were excluded from the subsequent analysis. The most common reasons for exclusion were: studies for meta-analyses applied inconsistent measurement methods and units of measurement, and in some cases, missing information regarding the sex of participants in the groups studied, and limited outcome data. The study selection process is summarized in Figure 1b.

#### 3.2.1. Quality Assessment of Included Studies for Biomarkers α-Synuclein and DJ-1

In accordance with the Cochrane Reviewers’ Handbook, the studies were assessed and graded to limit the risk of bias caused by inadequacies in study design, conduct, or analysis. In this case, each study was rated on three different levels according to the Newcastle–Ottawa Scale (NOS). Of the fourteen included studies, only one received the score “good quality”, while twelve investigations were of “fair quality”, and one study was considered of “low quality”. The results of the NOS scoring assessment are given in Table 6. Reasons for categorizing the studies as moderate quality were mostly due to the lack of sample size calculations, limitations in design (i.e., non-observance of confounding factors and lacking sound representativeness), or inaccuracy in elucidating the obtained data, such as disregarded non-respondents data.

#### 3.2.2. Compilation, Characteristics, and Outcome of Included Studies for Biomarker DJ-1

Five out of fourteen studies on saliva biomarkers for PD determined the levels of DJ-1 in this body fluid [28,29,30,32,36] (Table 4). The first study, published in 2011, quantified DJ-1 in saliva of 24 PD patients and 25 control group participants using the Western blot method (Becton Dickinson, Franklin Lakes, NJ, USA) [28]. They observed a moderate but non-significant increase in DJ-1 in the saliva of PD patients compared to healthy subjects (190 ± 70 ng/mL vs. 120 ± 30 ng/mL). In addition, DJ-1 levels did not correlate with UPDRS scores, i.e., DJ-1 levels were not influenced during the longitudinal course of PD. Three years later, the same group used Luminex assays to quantify DJ-1 levels in the buccal epithelium and in the cellular component of the very same saliva samples of the same case–control group as in their previous study [30]. In agreement with their previous results, DJ-1 levels were slightly but non-significantly increased in PD patients compared to unaffected controls (88 ± 8 pg/μg vs. 70 ± 8 pg/μg).

An investigation with two larger case–control groups also implemented Luminex assays (Bio-Rad; Abcam, Cambridge, UK) for determining DJ-1 levels in the saliva [29]. In the first case–control group (PD, *n* = 74; control, *n* = 12), they observed that salivary DJ-1 levels correlated with striatal dopaminergic function. Inspired by these results, they analyzed the saliva of an even larger case–control group (PD, *n* = 285; control, *n* = 91) for DJ-1 levels. They revealed that the DJ-I concentration in the saliva remained nearly unaltered when comparing PD patients and non-affected controls (4.11 ± 5.88 ng/mL vs. 3.86 ± 5,44 ng/mL). Underlining the results of Devic et al. [28], they further observed that DJ-1 levels did not correlate with UPDRS III scores.

Using ELISA (Thermo Fisher Scientific, Cramlington, UK) to detect DJ-1 in saliva of 16 PD patients and 22 healthy volunteers, Masters et al. observed that the concentration of DJ-1 was significantly increased in PD patients (0.84 µg/mL) comparing with controls (0.42 µg/mL) [32]. Notably, in this study normalized DJ-1 levels correlated with UPDRS scores, i.e., the longitudinal course of PD.

In 2018, Su et al. quantified the levels of DJ-1 in 27 PD sufferers and 27 control group participants using ELISA (Thermo Fisher Scientific) [36]. They observed a significant decrease in DJ-1 in the saliva of PD patients compared to healthy subjects (6.07 ± 3.23 ng/mL vs. 8.43 ± 4.33 ng/mL). Like Devic et al. and Kang et al., the DJ-1 levels were not related to UPDRS-II/III, HY stage, SS-12, MMSE, or MoCA of Parkinson’s disease group [28,29].

In sum, in three out of the five studies on DJ-1 levels in the saliva, DJ-1 levels were unaltered, or only slightly to moderately altered, but non-significantly elevated in PD patients compared to controls [28,29,30]. Masters et al. observed with a small case–control group a significant increase in DJ-1 [32]. In contrast, Su et al. showed a significant decrease in DJ-1 levels of PD sufferers [36]. Notably, studies published between 2019 and 2022 did not address the issues of DJ-1 in saliva of PD patients, being instead focused on the investigation of α-synuclein as a PD-relevant salivary biomarker.

#### 3.2.3. Compilation, Characteristics, and Outcome of Included Studies for Biomarker α-Synuclein

Twelve out of fourteen studies on saliva biomarkers for PD determined the levels of α-synuclein in this body fluid [28,30,31,33,34,35,36,37,39,40,41,42] (Table 4). In addition to DJ-1, Devic and colleagues quantified, in 2011, total α-synuclein levels in saliva of 24 PD patients and 25 control group participants using a Western blot method (Becton Dickinson, Franklin Lakes, NJ, USA) [28]. While markedly but non-significantly lower α-synuclein levels were found in PD sufferers (70 ± 80 pg/mL) than in the control group (110 ± 130 pg/mL), no significant difference for DJ-1 was observed. In addition, PD-associated depletion of α-synuclein correlated (although again non-significantly) with UPDRS scores, which reflects the longitudinal course of PD. In a follow-up study [30], the same group analyzed the cellular component of the saliva samples previously analyzed by Devic et al., (2011). Here, they observed a slight but non-significant increase in total α-synuclein levels in cellular saliva samples of PD patients compared to the unaffected control group.

In the same year, Al-Nimer and colleagues analyzed saliva samples from 20 PD patients and 20 control group participants for total salivary α-synuclein using the ELISA method (AnaSpec, Inc., Fremont, CA, USA) [31]. The results of their study show that there were significantly lower levels (65 ± 52.2 pg/mL) of total α-synuclein in the PD patients than in the control group (314.01 ± 435.9 pg/mL).

Two years later, saliva samples from 201 PD sufferers and 67 healthy volunteers were analyzed by means of Luminex assay (Bio-Rad, Abcam) for total and oligomeric α-synuclein, as well as the relationship between α-synuclein and α-synuclein SNP (single-nucleotide polymorphism) variants [33]. No significant difference was found between the levels of total α-synuclein in PD patients (128.66 ± 98.21 pg/mg) and the control group (131.31 ± 104.21 pg/mg). However, the study shows that the level of total α-synuclein can be manipulated by SNPs. In addition, although the numeric values of oligomeric α-synuclein were not reported by the authors, the study states that the ratio of oligomeric α-synuclein/total α-synuclein might be influenced by the disease stage of PD. In the early disease stage (Hoehn and Yahr Scale I), the ratio was significantly decreased in patients compared to controls, whereas in later disease stages (Hoehn and Yahr Scale II to IV), the ratio was significantly increased.

In 2016, Vivacqua et al. quantified the levels of total α-synuclein, oligomeric α-synuclein, and the ratio between these two biomarkers using ELISA (MyBioSource lab. Inc., San Diego, CA, USA) [34]. In this instance, a significant decrease in α-synuclein was detected in the saliva of 60 PD subjects (5.08 ± 3.01 pg/mL) compared to 40 healthy participants (31.3 ± 22.4 pg/mL), while the level of oligomeric α-synuclein was found to be significantly increased in PD patients (1.062 ± 0.266 ng/mL) compared, again, to the control group (0.498 ± 0.203 ng/mL). Consequently, a significant increase in the ratio of total and oligomeric α-synuclein was observed in PD patients (PD: 0.174 ± 0.044 vs. control: 0.065 ± 0.027). Three years later, the same authors conducted a larger scale follow-up study [39], including 112 PD patients and 80 participants in the control group. Again using ELISA (MyBioSource lab. Inc), the authors quantified the same biomarkers with comparable results, demonstrating a significant decrease in total α-synuclein in PD patients (7.104 ± 5.122 pg/mL) compared to the control group (28.444 ± 25.877 pg/mL), a significant elevation in oligomeric α-synuclein in PD sufferers (0.893 ± 1.949 ng/mL) compared to healthy subjects (0.217 ± 0.191 ng/mL), and a significantly elevated ratio between these biomarkers (PD: 0.235 ± 0.793 vs. control: 0.0126 ± 0.0079).

In an ELISA-supported investigation, total α-synuclein concentrations were moderately increased in saliva samples in PD patients (*n* = 115) compared to unaffected controls (*n* = 88) (PD: 285.42 ± 400.13 pg/mL; Control: 165.97 ± 272.3 pg/mL) [35]. However, the variance of the values was very large, and consequently, the increase was described as being non-significant. Interestingly, in this study, CSF samples were also analyzed, and, in CSF, total α-synuclein concentrations turned out to be significantly lower in PD patients compared to control subjects.

Su et al. quantified total α-synuclein concentrations in saliva samples of 27 PD patients and 27 healthy subjects using ELISA (Thermo Fisher Scientific) [36]. The concentration of total α-synuclein was found to be significantly lower in PD patients (1269.02 ± 16.09 pg/mL) compared to the control group (1350.51 ± 25.79 pg/mL). Similar to the beforementioned DJ-1 levels, salivary α-synuclein concentrations did not correlate with UPDRS-II/III, HY stage, SS-12, MMSE, or MoCA of the Parkinson’s disease group.

An electrochemiluminescence (ECL, no manufacturer details)-based study, using a case–control group of 74 PD patients and 60 healthy control subjects, demonstrated that the total α-synuclein levels turned out to unaltered (11.93 pg/mL vs. 12.23 pg/mL) [37]. However, they also showed significant elevated levels of oligomeric α-synuclein (7.03 pg/ng vs. 0.92 pg/ng), and the ratio of total and oligomeric α-synuclein (0.79 vs. 0.10) in saliva of PD patients compared to unaffected controls, confirming other observations [34,39].

The authors of a recently published study quantified the concentrations of total and oligomeric α-synuclein followed by the investigation of the ratio between these two biomarkers using ELISA (MyBioSource lab. Inc.) in 25 PD sufferers and 15 control group participants [40]. The concentration of total α-synuclein was found to be significantly lower in PD patients (159.4 ± 61.6 ng/mL) compared to the control group (229.9 ± 64 ng/mL), while oligomeric α-synuclein and the ratio of oligomeric α-synuclein/total α-synuclein were increased in PD patients (respectively, PD: 47.8 ± 11.8 ng/mL vs. control: 39.2 ± 9.2 ng/mL and PD: 0.35 ± 0.18 vs. control: 0.19 ± 0.08).

Chahine et al. recently analyzed biopsies of skin, colon, submandibular gland, CSF, serum, and saliva samples of total α-synuclein in 59 PD patients and 21 healthy volunteers using ELISA (no manufacturer details) [41]. While CSF samples demonstrated lower levels of α-synuclein in PD patients in comparison with the control group, its corresponding concentration in serum and saliva samples was not significantly different between PD patients (saliva, 65.6 ± 42.1 pg/mL) and healthy individuals (saliva, 64.4 ± 60.7 pg/mL).

One recently published investigation quantified saliva and serum samples of total α-synuclein, nitrotyrosinated proteins (3-NT-proteins), and the correlation between α-synuclein and 3-NT-proteins, in 45 PD patients and 30 healthy volunteers using ELISA (BlueGene Biotech Co., Ltd., Shanghai, China) [42]. Salivary concentration, as well as saliva and serum ratios of α-synuclein and 3-NT-proteins are similar in PD patients (α-synuclein in saliva, 361.9 ± 89 pg/mL) and the control group (α-synuclein in saliva, 372.1 ± 91pg/mL). Salivary α-synuclein and 3-NT-proteins did not correlate with any clinical feature. They also observed submandibular gland tissue in PD patients and detected nitrated α-synuclein, “Lewy-type” inclusions expressing 3-NT-α-synuclein. This recent discovery could be a viable technique for diagnosing PD.

To sum up, in twelve out of twelve studies on α-synuclein in the saliva, total α-synuclein levels have been determined [28,30,31,33,34,35,36,37,39,40,41,42]. In seven studies, total α-synuclein levels were found to be either significantly (five studies: [31,34,36,39,40] or markedly (two studies: [28,42]) reduced in PD patients compared to unaffected controls. In three further studies, the total α-synuclein levels remained unaltered between PD cases and controls [33,37,41], and, in two studies, the total α-synuclein levels increased [30,35]. Therefore, it is questionable if total α-synuclein is a robust saliva biomarker for PD.

In five out of twelve studies, on α-synuclein in the saliva [33,34,37,39,40], oligomeric α-synuclein levels have been measured in addition to total α-synuclein. In four studies, oligomeric α-synuclein significantly accumulated in the saliva of PD patients compared to controls, and consequently, the ratio of oligomeric α-synuclein/total α-synuclein significantly increased as well [34,37,39,40]. In the fifth study, the significant increase in the ratio of oligomeric α-synuclein/total α-synuclein was restricted to advanced PD stages [33]. In total, measuring oligomeric α-synuclein and determining the ratio of oligomeric α-synuclein/total α-synuclein turned out to be favorable over determining total α-synuclein levels only.

## 4. Discussion

The aim of the present work was to review the currently available literature regarding the diagnostic potential of the biomarkers amyloid-β, tau, and α-synuclein and DJ-1 in the saliva of patients suffering from AD and PD, respectively (Table 7). In this process, ten quantitative studies for AD and fourteen quantitative studies for PD were included in this systematic review, indicating that the number of studies that met the inclusion criteria is limited. Although a brief analysis of relevant studies indicated no serious concerns regarding testing methods, sample size, or target group structure, a more detailed analysis revealed several shortcomings, including low methodological quality, diversity, and heterogeneity. Indeed, the present research was planned to become a meta-analysis; however, due to the lack of homogeneous data, the authors of the present work decided to transform their findings into a systematic review (which was in accordance with a recommendation of the Cochrane Center Austria, Krems, Austria). Consequently, the essential characteristics of the reviewed studies should take the prominent stage in the discussion that follows.

In general, the observations of the present systematic review suggest that Aβ42 could be a promising AD-relevant salivary biomarker in the future. Although almost all studies included in the review suffered from a small sample size, five investigations have shown a similar trend and reported on higher salivary concentration of Aβ42 in AD patients [18,20,22,23,24], while two studies failed to quantify this biomarker in saliva [19,21] and one study showed a reduced salivary concentration of Aβ42 in AD patients [27]. Four studies considered for, lately unsuccessful, quantitative comparison [18,22,23,24] showed, with I^2^ = 98%, high statistical heterogeneity, which might be due to variables such as different clinical diagnostic criteria for AD, inclusion of patients at different stages of disease, differences in collection and storage of saliva samples, and the measurement methods of the biomarkers. Since three studies [18,20,27] aiming to investigate the concentration of Aβ40 in saliva found no statistical differences between AD patients and healthy individuals, Aβ40 seems to represent a more unreliable salivary biomarker relative to Aβ42. However, in relation to Aβ42, Aβ40 might be potentially useful as a reference marker, thus, the investigation of the Aβ42/Aβ40 ratio could provide more reliable results in a large-scale case–control study, which is still missing.

Salivary t-tau does not appear suitable for valid diagnosis of Morbus Alzheimer. In previous studies, in which t-tau values were shown [19,21,26,27], t-tau consistently remained unaltered in AD patients compared to controls. In contrast, p-tau and the p-tau/t-tau ratio markedly or significantly increased in several studies [19,21,25]. In only one study, p-tau non-significantly decreased [27]. However, sufficient data extraction was limited due to the lack of numeric data. Further challenges, hindering attempts of quantitative analysis, became obvious. In all five studies, the measurements of salivary t-tau or p-tau were conducted by means of five, indeed comparable, but nevertheless different, methods: ELISA, Luminex, SIMOA, MILLIPLEX^®^ MAP, and Western blot, which probably demonstrates low sensitivity in assessing small concentrations of the studied biomarker.

The protein α-synuclein is considered a specific biomarker for PD, thus, twelve studies included in the present systematic review investigated different α-synuclein types in saliva, namely total α-synuclein, the oligomeric α-synuclein, and the ratio between these two biomarkers. In seven studies, total α-synuclein levels were depleted in PD patients compared to controls [28,31,34,36,39,40,42]. In other studies, the total α-synuclein levels remained unaltered [33,37,41], and in two studies the total α-synuclein levels even increased [30,35], proposing total α-synuclein being an unreliable salivary biomarker. In contrast, oligomeric α-synuclein and the ratio of oligomeric α-synuclein/total α-synuclein increased in all studies in PD patients compared to controls, at least at later PD stages [33,34,37,39,40]. Thus, measuring oligomeric α-synuclein and determining the ratio of oligomeric α-synuclein/total α-synuclein might be a promising salivary biomarker for PD. Apart from similar limitations mentioned as reasons for failure of analytic approach in case of Aβ42, additional challenges became evident. Thus, of particular note are the results of the recent study, demonstrating considerably deviating concentration of total and oligomeric α-synuclein in contrast to the other included investigations. Most authors presented the concentrations of α-synuclein in pg/mL, there were, however, some researchers, who reported on their results using varying scale units, such as ng/mL, pg/µg, pg/ng, pg/mL or ng/mL. To allow for comparison and to reach more conclusiveness, all concentration related measurements were converted into equal units (pg/mL). While the majority of units revealed, therefore, plausible concentration ranges, only the study mentioned above demonstrated, then using converted units, unphysiologically high concentrations of α-synuclein (with means between 159,400 and 229,900 pg/mL), independent of case or control. Interestingly, if one assumes that the α-synuclein concentrations in this study would be given in pg/mL, the result would be in agreement with all other investigations conducted in this field. However, these thoughts are speculative in nature, since the corresponding authors of the respective paper did not respond to our repeated queries regarding the possible drawback described above.

The DJ-1 protein is considered a second potential specific salivary biomarker for the diagnosis of Morbus Parkinson; therefore, four studies aimed to quantify DJ-1 in saliva of PD patients, and to compare it with the respective concentrations in controls. In three out of the five studies on DJ-1 levels in the saliva, DJ-1 levels were unaltered, or only slightly to moderately altered, but non-significantly elevated in PD patients compared to controls [28,29,30]. Only one study showed a significant decrease in DJ-1 levels in saliva [36], and another one with a small case–control group showed a significant accumulation of DJ-1 [32]. Here, it was demonstrated that the saliva composition of PD patients generally seems to be different than in healthy individuals, emphasizing careful handling of DJ-1 as a stand-alone biomarker for PD diagnosis, since the increase in DJ-1 was observed along with the simultaneous increases in albumin, amylase, and total protein. In particular, the latter fact leads to the assumption that further, yet not discussed, biomarkers or other substances contained in saliva might be specific as well for Morbus Alzheimer as for Morbus Parkinson. Thus, in connection with the diagnosis Morbus Alzheimer and Morbus Parkinson, investigation concerning the potential diagnostic significance of salivary substances, such as acetylcholinesterase (AChE), lactoferrin, cortisol, trehalose IgA-, and glucose-levels, as well as different metabolites, have been conducted. However, due to the inadequate number of studies featured in the present study, as well as the small sample sizes and non-standardized measurement methods, it is currently not possible to make a precise statement about the possibility of using them as potential specific diagnostic tools. Other potential salivary biomarkers may arise in the future, such as elastin-like or elastin-derived peptides (ELPs/EDPs) for AD. These species are found in body fluids, and have been observed to initiate or facilitate AD progression [43,44,45,46].

Overall, it is apparent that saliva samples do indeed represent an interesting and non-invasive method to diagnose neurodegenerative diseases such as AD and PD. Although ELISA was able to detect and to quantify several salivary biomarkers, its capability and accuracy to detect same substances in CSF samples currently still seem be superior due to the lack of standardized and validated protocols for investigations of saliva. Hence, there is a need to develop standardized measurement methods and to conduct studies examining larger population cohorts. Notwithstanding, these promising attempts, which might contribute to substantial relief from invasive diagnostic approaches, should justify and encourage the interest in planning and conducting more high-quality studies.

## 5. Conclusions

Summarizing the currently available studies, conducted between 2010 and 2022, and concerning the applicability of salivary biomarkers amyloid-β, tau, α-synuclein, and DJ-1 for the diagnosis of either AD or PD, the authors of the present investigation cautiously point to the potential of saliva as a non-invasive biomarker source. Although further investigations to determine the reliability of such biomarkers in detecting disease pathology or monitoring of its progression are clearly needed, the current state of research indicates that salivary biomarker Aβ42 would be the most promising for the diagnosis of Morbus Alzheimer. Despite currently unsatisfactory evidence, the determination of total α-synuclein, oligomeric α-synuclein, and their ratio, in saliva of patients suffering from Morbus Parkinson would also be of interest.

## Figures and Tables

**Figure 1 biomedicines-10-01702-f001:**
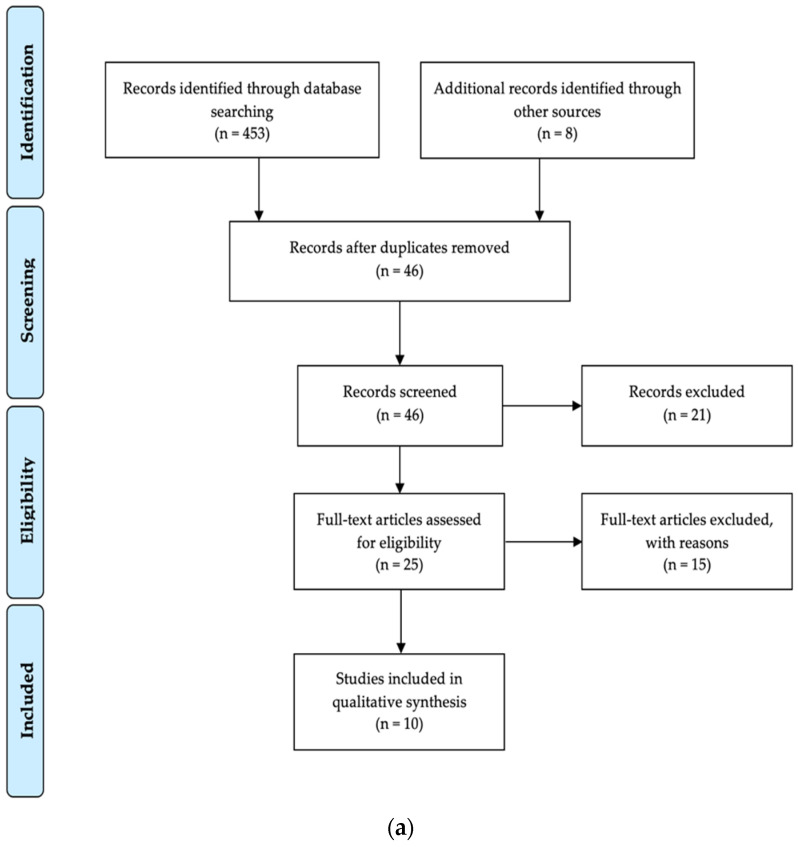
(**a**) PRISMA flow diagram showing the number of studies concerning salivary biomarkers for AD that were identified, screened, assessed for eligibility, excluded, and included in the systematic research. (**b**) PRISMA flow diagram showing the number of studies concerning salivary biomarkers for PD that were identified, screened, assessed for eligibility, excluded, and included in the systematic research.

**Table 1 biomedicines-10-01702-t001:** Population characteristics in studies on salivary biomarkers for AD.

Study (Author/Year)	Clinical Diagnosis	Methods	Population Characteristics (Population Size, Sex, Age (Mean/Range))
1. Bermejo-Pareja et al., 2010 [18]	AD characterized using DSM-IV and NINCDS-ADRDA; vascular dementia excluded using DSM-III-R	MMSE MRT and/or CT Extensive biochemical measurements ApoE genotyping	AD (*n* = 70): Sex: m 21/f 49 Age: 77.20 years (60–91) Disease duration: 2.56 years (0–12) MMSE: 17 (4–28) *Mild AD* (*n* = *29*) *Moderate AD* (*n* = *24*) *Severe AD* (*n* = *17*) Control (*n* = 56): Sex: m 17/f 39 Age: 74.35 years (64–85) Sex-, age-, and ethnicity-matched control
2. Shi et al., 2011 [19]	AD characterized using NINDS-ADRDA	MMSE CDRS	AD (*n* = 21): Sex: m 10/f 11 Age: 68.80 years (52–85) Disease duration: 4.4 years (2–10) CDRS: 0.05 (0–0.5) MMSE: 19.2 (4–29) Control (*n* = 38): Sex: m 19/f 19 Age: 69.00 years (40–88) CDRS: 1.05 (0.5–2) MMSE: 29.4 (27–30)
3. Kim et al., 2014 [20]	NS	MMSE	AD (*n* = 28): Sex, age, and ethnicity NS *Mild AD* (*MCI*) (*n* = *NS*) *Severe AD* (*n* = *NS*) Control (*n* = 17): Sex, age, and ethnicity NS
4. Lau et al., 2015 [21]	NS	MMSE CDR-SOB	AD (*n* = 20): Sex: m 8/f 12 Age: 72.50 ± 7.68 years MMSE: 18.15 ± 5.4 CDR-SOB: 6.25 ± 2.67 Control (*n* = 20): Sex: m 9/f 11 Age: 66.10 ± 7.79 years MMSE: 28.7 ± 1.11 CDR-SOB: 0.23 ± 0.25
5. Lee et al., 2017 [22]	NS	NS	AD (*n* = 7): Sex: m 3/f 4 Age: 76.86 years (57–86) AD family history (*n* = 3) Pre-AD (*n* = 3): Sex: m 0/f 3 Age: 54.33 years (51–60) AD family history (*n* = 3) Control (*n* = 26): Sex: m 17/f 9 Age: 54.62 years (19–92) AD family history (*n* = 9)
6. McGeer et al., 2018 [23]	NS	NS	AD (*n* = 23): Sex: m 8/f 15 Age: 74.14 ± 11.31 years Control (*n* = 31): Sex: m 20/f 11 Age: 57.06 ± 21.73 years *High control* (*n* = *6*)*:* (predicted risk for AD) Sex: m 3/f 3 Age: 69.00 ± 8.97 years *Low control* (*n* = *25*)*:* Sex: m 17/f 8 Age: 54.20 ± 23.0 years
7. Sabbagh et al., 2018 [24]	NIA-AA Exclusion of subjects with medical history of major systemic diseases that affect cognitive function	MMSE	AD (*n* = 15): Sex: m 7/f 8 Age: 77.8 ± 1.8 years MMSE: 19.0 ± 1.3 Control (*n* = 7): Sex: m 2/f 5 Age: 60.4 ± 4.7 years MMSE: 29.0 ± 0.4
8. Pekeles et al., 2019 [25]	NIA-AA	Clock-drawing test MoCA WMS	AD (*n* = 46): Sex: m 24/f 22 Age: 80 years (median), 9 (IQR) MCI (*n* = 55): Sex: m 23/32 Age: 78 years (median), 14 (IQR) Control (*n* = 47): Sex: m 15/f 32 Age: 73 years (median), 6 (IQR) MoCA ≥ 25
9. Ashton et al., 2018 [26]	NS	CDR MMSE MRI *APOE* genotyping	AD (*n* = 53): Sex: m 23/f 30 Age: 81.4 ± 6.6 years CDR: 0.89 ± 0.82 MMSE: 22.3 ± 5.7 MCI (*n* = 68): Sex: m 33/f 35 Age: 79.8 ± 7.4 years CDR: 0.48 ± 0.14 MMSE: 26.8 ± 2.3 Control (*n* = 160): Sex: m 66/f 94 Age: 78.0 ± 6.7 years CDR: 0.15 ± 0.24 MMSE: 28.9 ± 1.1
10. Tvarijonaviciute et al., 2020 [27]	AD characterized using GDS	Controls characterized using MMSE	AD (*n* = 69): Sex: NS Age: 75.6 ± 7.2 years MMSE: NS Control (*n* = 83): Sex: NS Age: 75.6 ± 7.2 years MMSE: ≥ 28

CDRS: Clinical Dementia Rating Scale; CDR-SOB: Clinical Dementia Rating-Sum of Boxes (CDR-SOB, score range from 0, cognitive normality, to 18, maximal cognitive impairment); CT: computed tomography scan; GDS: Global Deterioration Scale of Reisberg; IQR: interquartile range; MCI: mild cognitive impairment; MMSE: mini-mental state examination (MMSE score range from 0, severe impairment, to 30, no impairment); MoCA: Montreal Cognitive Assessment (maximal score of 30); MRI: magnetic resonance imaging; NIA-AA: AD criteria established by the National Institute on Aging and the Alzheimer’s Association; NS: not specified; WMS: Wechsler Memory Scale (with Logical Memory 2 score). Population main groups are underlined. Population subgroups are in italics.

**Table 2 biomedicines-10-01702-t002:** Salivary biomarkers for AD.

Study (Author/Year)	Biomarker Tested	Biomarker/ Concentration (Saliva)	Detection Method (Saliva)	Body Fluids Tested
1. Bermejo- Pareja et al., 2010 [18]	Aβ42	Aβ42 markedly elevated in AD patients compared to controls: AD: 6.81 ± 20.04 pg/mL Mild AD: 7.67 ± 16.25 pg/mL Moderate AD: 11.70 ± 34.76 pg/mL Severe AD: 3.03 ± 3.49 pg/mL Control: 2.89 ± 4.96 pg/mL No significant changes with Aβ42 in plasma	ELISA	Saliva Plasma
	Aβ40	Aβ40 unaltered in AD patients compared to controls: AD: 22.3 ± 4.88 pg/mL Mild AD: 21.87 ± 5.7 pg/mL Moderate AD: 21.5 ± 4.17 pg/mL Severe AD: 23.92 ± 4.55 pg/mL Control: 20.82 ± 5.55 pg/mL No significant changes with Aβ40 in plasma	ELISA	Saliva Plasma
	Aβ42/ Aβ40 ratio	Aβ42/Aβ40 ratio is moderately (but non-significantly) elevated in patients with mild and moderate AD compared to controls	ELISA	Saliva Plasma
2. Shi et al., 2011 [19]	Aβ42	Data cannot be extracted Aβ42 not detected in the cohort, neither with Luminex nor with IP/MS	Luminex IP/MS	Saliva
	t-tau	Data cannot be extracted tau detected with IP/MS in saliva t-tau unaltered in patients with AD compared to controls	Luminex IP/MS	Saliva
	p-tau	Data cannot be extracted p-tau markedly elevated in patients with AD compared to controls	Luminex	Saliva
	p-tau/ t-tau ratio	Data cannot be extracted p-tau/t-tau ratio significantly elevated in patients with AD compared to controls	Luminex	Saliva
3. Kim et al., 2014 [20]	Aβ42	Data cannot be extracted Aβ42 significantly elevated in patients with severe AD compared to patients with mild AD or controls	MIA ELISA	Saliva
	Aβ40	Data cannot be extracted Aβ40 non-significantly elevated in patients with severe AD compared to patients with mild AD or controls	MIA ELISA	Saliva
	Aβ42/Aβ40 ratio	Data cannot be extracted Aβ42/Aβ40 ratio is not elevated in patients with severe or mild AD compared to controls Aβ42/Aβ40 ratio is elevated in patients with severe AD compared to patients with mild AD (significance unclear)	MIA	Saliva
4. Lau et al., 2015 [21]	Aβ42	Data cannot be extracted Aβ42 not detected in salivary samples of patients with AD or controls	ELISA	Saliva
	t-tau	Data cannot be extracted t-tau unaltered in patients with AD compared to controls	ELISA	Saliva
	p-tau	Data cannot be extracted p-tau moderately (but non-significantly) elevated in patients with AD compared to controls	ELISA	Saliva
	p-tau/ t-tau ratio	Data cannot be extracted p-tau/t-tau ratio moderately (but non-significantly) elevated in patients with AD compared to controls	ELISA	Saliva
5. Lee et al., 2017 [22]	Aβ42	Aβ42 significantly elevated in AD patients compared to controls Aβ42 markedly elevated in pre-AD patients compared to controls Aβ42 unaltered in AD patients compared to pre-AD patients AD: 59.07 ± 6.33 pg/mL Pre-AD: 56.14 ± 7.12 pg/mL Control: 22.06 ± 0.41 pg/ml	ELISA	Saliva
6. McGeer et al., 2018 [23]	Aβ42	Aβ42 significantly elevated in AD patients compared to controls and low controls Aβ42 unaltered in AD patients compared to high controls AD: 53.95 ± 2.24 pg/mL Control: 26.55 ± 1.85 pg/mL High control: 45.96 ± 3.01 pg/mL Low control: 21.54 ± 0.19 pg/mL	ELISA	Saliva
7. Sabbagh et al., 2018 [24]	Aβ42	Aβ42 significantly elevated in AD patients compared to controls AD: 51.7 ± 1.6 pg/mL Control: 21.1 ± 0.3 pg/mL	ELISA	Saliva
8. Pekeles et al., 2019 [25]	p-tau/ t-tau ratio	Data cannot be extracted p-tau/t-tau ratio significantly elevated in patients with AD compared to controls (for three of the four phosphorylation sites tested)	Western blot (for saliva) ELISA (for CSF)	Saliva CSF
9. Ashton et al., 2018 [26]	t-tau	t-tau non-significantly altered among AD, MCI, and controls AD: 12.3 ng/L MCI: 9.8 ng/L Control: 9.6 ng/L	SIMOA immunoassay	Saliva
10. Tvarijonaviciute et al., 2020 [27]	Aβ40	Aβ40 non-significantly elevated in AD patients compared to controls AD: 21.98 ± 16.94 pg/mL Control: 19.97 ± 6.35 pg/mL	MILLIPLEX^®^ MAP	Saliva
	Aβ42	Aβ42 non-significantly decreased in AD patients compared to controls AD: 3.15 ± 0.72 pg/mL Control: 3.57 ± 0.93 pg/mL	MILLIPLEX^®^ MAP	Saliva
	t-tau	t-tau unaltered in patients with AD compared to controls AD: 21.57 ± 22.11 pg/mL Control: 21.15 ± 16.58 pg/mL	MILLIPLEX^®^ MAP	Saliva
	p-tau	p-tau moderately (but non-significantly) decreased in patients with AD compared to controls AD: 40.33 ± 42.95 pg/mL Control: 42.5 ± 38.35 pg/ml	MILLIPLEX^®^ MAP	Saliva

CSF: cerebrospinal fluid; ELISA: enzyme-linked immunosorbent assay; IP: immunoprecipitation; MIA: magnetic immunoassay; MS: mass spectrometry; NS: not specified. Biomarker concentrations of main population groups are underlined.

**Table 3 biomedicines-10-01702-t003:** Population characteristics in studies on salivary biomarkers for PD.

Study (Author/Year)	Clinical Diagnosis	Methods	Population Characteristics (Population Size, Sex, Age (Mean/Range))
1. Devic et al., 2011 [28]	UK PD Society Brain Bank clinical diagnostic criteria for PD as determined by a movement disorder specialist	UPDRS	PD (*n* = 24): Sex: m 17/f 7 Age: 63.5 ± 11.3 years Duration of disease: 8.5 ± 6.4 years Patients with UPDRS III scores: 18/24 Control (*n* = 25): Sex: m 11/f 14 Age: 58.0 ± 10.4 years
2a. Kang et al., 2014 (pilot study) [29]	UK PD Society Brain Bank clinical diagnostic criteria for PD as determined by at least two senior movement disorder specialists	HAMD-17 MMSE REM Sleep Behavior Disorder Scale SCOPA-AUT UPDRS III HY Scale	PD (*n* = 74): Sex: m 50/f 24 Age: 61.8 ± 7.8 years Duration of disease: 4.36 ± 3.59 years UPDRS III: 17.63 ± 11.69 Control (*n* = 12): Sex: m 6/f 6 Age: 55.5 ± 6.11 years
2b. Kang et al., 2014 (large cohort study) [29]	UK PD Society Brain Bank clinical diagnostic criteria for PD as determined by at least two senior movement disorder specialists	HAMD-17 MMSE REM Sleep Behavior Disorder Scale SCOPA-AUT UPDRS III HY Scale	PD (*n* = 285): Sex: m 171/f 114 Age: 63.34 ± 9.11 years UPDRS III: 23.8 ± 15.7 Control (*n* = 91): Sex: m 59/f 32 Age: 61.59 ± 10.61 years
3. Stewart et al., 2014 [30]	UK PD Society Brain Bank clinical diagnostic criteria for PD as determined by a movement disorder specialist	UPDRS	PD (*n* = 24): Sex: m 17/f 7 Age: 63.5 ± 11.3 years Duration of disease: 8.5 ± 6.4 years Patients with UPDRS III scores: 18/24 (see Devic et al., 2011) Control (*n* = 25): Sex: m 11/f 14 Age: 58.0 ± 10.4 years (see Devic et al., 2011)
4. Al-Nimer et al., 2014 [31]	UK PD Society Brain Bank clinical diagnostic criteria for PD	MDS-UPDRS	PD (*n* = 20): Sex: m 16/f 4 Age: 64.4 ± 10.6 years Duration of disease: 6.55 ± 6.83 years Family history: 6 Control (*n* = 20): Sex: m 18/f 2 Age: 65.4 ± 8.2 years
5. Masters et al., 2015 [32]	Queen Square Brain Bank clinical diagnostic criteria for PD as determined by two trained clinicians	MDS-UPDRS ACB	PD (*n* = 16) Control (*n* = 22) (further details not available)
6. Kang et al., 2016 [33]	UK PD Society Brain Bank clinical diagnostic criteria for PD	UPDRS III Genotyping for SNCA variants	PD (*n* = 201): Sex: m 122/f 79 Age: 63.18 ± 9.67 years Control (*n* = 67): Sex: m 41/f 26 Age: 61.04 ± 10.01 years
7. Vivacqua et al., 2016 [34]	Queen Square Brain Bank clinical diagnostic criteria for PD	BDI-II HY Scale MDS-UPDRS MoCA FAB LEDD	PD (*n* = 60): Sex: m 31/f 29 Age: 66.3 ± 8.78 years MoCA score > 18 FAB score > 12 Control (*n* = 40): Sex: m 22/f 18 Age: 68.3 ± 7.9 years
8. Goldman et al., 2018 [35]	UK PD Society Brain Bank clinical diagnostic criteria for PD; atypical or secondary parkinsonian syndromes excluded	HY Scale MDS-UPDRS MoCA	PD (*n* = 115): Sex: m 72/f 43 Age: 68.24 ± 6.40 years Duration of disease: 8.34 ± 3.09 years HY stage: 2.18 ± 0.67 MoCA score: 26.76 ± 2.56 UPDRS III score: 39.13 ± 13.19 Control (*n* = 88): Sex: m 59/f 29 Age: 65.64 ± 7.36 years MoCA score ≥ 26 No first-degree PD family members
9. Su et al., 2018 [36]	International Parkinson and Movement Disorder Society (The Movement Disorder Society)	HY Scale MDP-UPDRS-II/III SS-12 MMSE MoCA	PD (*n* = 27): Sex: m 15/f 12 Age: 61.52 ± 9.57 years HY stage: 1–3 UPDRS II score: 10.93 ± 5.35 UPDRS III score: 25.52 ± 11.34 SS-12: 4.70 ± 2.89 Control (*n* = 27): Sex: m 15/f 12 Age: 58.37 ± 9.85 years
10a. Cao et al., 2019 [37]	UK PD Society Brain Bank clinical diagnostic criteria for PD as diagnosed by two expert professional neurologists	HY Scale UPDRS-III	PD (*n* = 74): Sex: m 40 /f 34 Age: 59.62 ± 8.57 years HY stage: 2.5 (2–3) UDPRS-III: 38.40 ± 19.39 Control (*n* = 60): Sex: m 26/f 34 Age: 58.75 ± 9.85 years
10b. Cao et al., 2020 [38]	UK PD Society Brain Bank clinical diagnostic criteria for PD as diagnosed by two expert professional neurologists	HY Scale HAMA HAMD MMSE MoCA RBDSQ	PD (*n* = 26): Sex: m 12/f 14 Age: 57.31 ± 7.78 years Duration of disease: 2.64 ± 1.19 years HY stage: 2.50 ± 0.62 UPDRS III score: 40.77 ± 16.00 MSA-P (*n* = 16): Sex: m 9/f 7 Age: 56.82 ± 6.45 years Duration of disease: 3.06 ± 1.73 years HY stage: 2.81 ± 0.75 UPDRS III score: 42.75 ± 18.87
11. Vivacqua et al., 2019 [39]	Queen Square Brain Bank clinical diagnostic criteria for PD as determined by two trained clinicians	FAB HY Scale LEDD MDS-UPDRS MoCA PSPRS	PD (*n* = 112): Sex: m 59/f 53 Age: 69.01 ± 11.16 years Duration of disease: 6.29 ± 5.03 years FAB: 16.376 ± 1.918 HY stage: 2.11 ± 0.74 MDS-UPDRS score: 38.06 ± 21.06 MoCA: 26.60 ± 3.284 PSP (*n* = 22): Sex: m 12/10 Age: 68.84 ± 6.16 years Duration of disease: 3.07 ± 1.31 years FAB: 14.153 ± 1.918 HY stage: 3.19 ± 0.15 MDS-UPDRS score: 32.384 ± 11.2 MoCA: 22.538 ± 3.9 PSPRS: 32.61 ± 11.5 Control (*n* = 90): Sex: m 53/f 37 Age: 62.09 ± 15.08 years
12. Shaheen et al., 2020 [40]	UK PD Society Brain Bank clinical diagnostic criteria for PD	BRS HY Scale LEDD UPDRS	PD (*n* = 25): Sex: m 15/f 10 Age: 60.1 ± 5.6 years Duration of disease: 0.5 to 10.0 years UPDRS III score: 29.9 ± 11.1 HY stage: 2.08 ± 0.6 Control (*n* = 15): Sex: m 10/f 5; Age: 60.0 ± 6.7 years Age- and sex-matched control
13. Chahine et al., 2020 [41]	NS	HY Scale MDS-UPDRS MoCA RBDSQR SCOPA-AUT	PD (*n* = 59): Sex: m 41/f 18 Age: 63.1 ± 8.6 years Duration of disease: 4.81 ± 4.58 UPDRS III score: 26.4 ± 11.9 Early PD (*n* = 18) Moderate PD (*n* = 20) Advanced PD (*n* = 21) Control (*n* = 21): Sex: m 9/f 12 Age: 61.0 ± 6.3 years UPDRS III score: 1.1 ± 2.3
14. Fernández-Espejo et al., 2021 [42]	UK PD Society Brain Bank clinical diagnostic criteria for PD SPECT scans	HY Scale MDS-UPDRS Modified Schwab and England ADL	PD (*n* = 45): Sex: m 27/f 18 Age: 61.4 ± 18.5 years Duration of disease: 9.9 ± 6.8 HY stage: 2.1 ± 0.8 Modified Schwab and England ADL: 86 ± 25 MDS-UPDRS III score: 24 ± 12 MDS-UPDRS IV score: 1.2 ± 2.4 MDS-UPDRS (I-III)) score: 37.2 ± 20 Control (*n* = 30): Sex: m 18/f 12 Age: 59.6 ± 11 years

ACB: Anti-Cholinergic Burden Score; ADL: Modified Schwab and England Activities of Daily Living; BDI-II: Beck Depression Inventory; BRS: Bradykinesia Rigidity Score; FAB: Frontal Assessment Battery; HAMA: Hamilton Anxiety Rating Scale; HAMD-17: 17-item Hamilton Rating Scale for Depression; HY: Hoehn and Yahr Scale; LEDD: L-Dopa equivalent daily dose; MDS-UPDRS: Movement Disorder Society-revision of Unified Parkinson’s Disease Rating Scale; MMSE: mini-mental state examination; MoCA: Montreal Cognitive Assessment; NS: not specified; PSP: Progressive Supranuclear Palsy; PSPRS: PSP Rating Scale; RBDSQ: Rapid Eye Movement Sleep Behavior Disorder Screening Questionnaire; SCOPA-AUT: Scales for Outcome in Parkinson’s disease—Autonomic dysfunction; SS-12: Sniffin’ Sticks Screening 12 Test; UPDRS: Unified Parkinson’s Disease Rating Scale. Population main groups are underlined.

**Table 4 biomedicines-10-01702-t004:** Salivary biomarkers for PD.

Study (Author/Year)	Biomarker Tested	Biomarker/ Concentration (Saliva)	Detection Method (Saliva)	Body Fluids Tested
1. Devic et al., 2011 [28]	Total α-Syn	Total α-Syn markedly (but non-significantly) reduced in PD compared to controls: PD: 70 ± 80 pg/mL Control: 110 ± 130 pg/mL α-Syn depletion markedly correlated (but non-significantly) with UPDRS scores (i.e., disease severity)	Luminex IP/Western blot	Saliva
	DJ-1	DJ-1 moderately (but non-significantly) increased in PD compared to controls: PD: 190 ± 70 ng/mL Control: 120 ± 30 ng/mL DJ-1 enrichment did not correlate with UPDRS scores (i.e., disease severity)	Luminex IP/Western blot	Saliva
2a. Kang et al., 2014 (pilot study) [29]	DJ-1	Moderate correlation between salivary DJ-1 levels and striatal dopaminergic function	Luminex	Saliva
2b. Kang et al., 2014 (large cohort study) [29]	DJ-1	DJ-1 levels were unaltered in PD patients compared to controls: PD: 4.11 ± 5.88 ng/mL Control: 3.86 ± 5.44 ng/mL DJ-1 enrichment did not correlate with UPDRS III scores (i.e., disease severity)	Luminex	Saliva
3. Stewart et al., 2014 [30]	Total α-Syn	Total α-Syn slightly (but non-significantly) increased in cellular components of saliva in PD patients compared to controls (data extracted from paper graph): PD: 0.42 ± 0.09 pg/μg Control: 0.36 ± 0.03 pg/μg	Luminex	Cellular components of saliva
	DJ-1	DJ-1 levels were unaltered in cellular components of saliva in PD patients vs. controls (data extracted from paper graph): PD: 88 ± 8 pg/μg Control: 70 ± 8 pg/μg	Luminex	Cellular components of saliva
4. Al-Nimer et al., 2014 [31]	Total α-Syn	Total α-Syn significantly reduced in PD compared to controls: PD 65 ± 52.2 pg/mL Control: 314.01 ± 435.9 pg/mL	ELISA	Saliva
5. Masters et al., 2015 [32]	DJ-1	DJ-1 levels significantly elevated in PD patients compared to controls: PD: 0.84 µg/mL Control: 0.42 µg/mL After normalization for total protein concentration, no alteration of DJ-1 in PD patients compared to controls Normalized DJ-1 levels correlated with UPDRS scores (i.e., disease severity)	Western blot	Saliva
6. Kang et al., 2016 [33]	Total α-Syn	Total α-Syn levels unaltered in PD patients compared to controls: PD: 128.66 ± 98.21 pg/mg Control: 131.31 ± 104.2 pg/mg	Luminex	Saliva
	Oligomeric α-Syn/ total α-Syn ratio	Oligomeric α-Syn/total α-Syn significantly decreased in early disease state (HY I), but significantly increased in later disease states (HY II to IV)	Western blot after size exclusion chromatography	Saliva
7. Vivacqua et al., 2016 [34]	Total α-Syn	Total α-Syn significantly reduced in PD patients compared to controls: PD: 5.08 ± 3.01 pg/mL Control: 31.3 ± 22.4 pg/mL	ELISA	Saliva
	Oligomeric α-Syn	Oligomeric α-Syn significantly increased in PD patients compared to controls: PD: 1.062 ± 0.266 ng/mL Control: 0.498 ± 0.203 ng/mL	ELISA	Saliva
	Oligomeric α-Syn/ total α-Syn ratio	Oligomeric α-Syn/total α-Syn ratio significantly increased in PD patients compared to controls: PD: 0.174 ± 0.044 Control: 0.065 ± 0.027	ELISA	Saliva
8. Goldman et al., 2018 [35]	Total α-Syn	Total α-Syn moderately (but non-significantly) increased in PD patients compared to controls: PD: 285.42 ± 400.13 pg/mL Control: 165.97 ± 272.3 pg/mL	ELISA	Saliva CSF Plasma
9. Su et al., 2018 [36]	Total α-Syn	Total α-Syn significantly reduced in PD patients compared to controls: PD: 1269.02 ± 16.09 pg/mL Control: 1350.51 ± 25.79 pg/mL	ELISA	Saliva
	DJ-1	DJ-1 levels significantly decreased in PD patients compared to controls: PD: 6.07 ± 3.23 ng/mL Control: 8.43 ± 4.33 ng/mL	ELISA	Saliva
10.a Cao et al., 2019 [37]	Total α-Syn	Total α-Syn in PD patients unaltered compared to controls: PD: 11.93 (6.23~28.11) pg/ng Control: 12.23 (5.47~58.83) pg/ng (mean and interquartile range)	Extracellular Vesicle Enrichment Kit followed by ECL immunoassays	Extracellular vesicles in saliva
	Oligomeric α-Syn	Oligomeric α-Syn in PD patients significantly increased compared to controls: PD: 7.03 (3.58~12.11) pg/ng Control: 0.92 (0.49~1.61) pg/ng (mean and interquartile range)	Extracellular Vesicle Enrichment Kit followed by ECL immunoassays	Extracellular vesicles in saliva
	Oligomeric α-Syn/ total α-Syn ratio	Oligomeric α-Syn/total α-Syn ratio significantly increased in PD patients compared to controls: PD: 0.79 (0.23~1.82) Control: 0.10 (0.04~0.28) (mean and interquartile range)	Extracellular Vesicle Enrichment Kit followed by ECL immunoassays	Extracellular vesicles in saliva
10b. Cao et al., 2020 [38]	Total α-Syn	Total α-Syn increased in PD patients compared to MSA-P patients: PD: 8.07 ± 4.71 pg/ng MSA-P: 5.44 ± 1.50 pg/ng	Extracellular Vesicle Enrichment Kit followed by ECL immunoassays	Extracellular vesicles in saliva
	Oligomeric α-Syn	Oligomeric α-Syn unaltered in PD patients compared to MSA-P patients: PD: 8.25 ± 3.98 pg/ng MSA-P: 7.29 ± 4.44 pg/ng	Extracellular Vesicle Enrichment Kit followed by ECL immunoassays	Extracellular vesicles in saliva
11. Vivacqua et al., 2019 [39]	Total α-Syn	Total α-Syn significantly reduced in PD patients compared to control and to PSP: PD: 7.104 ± 5.122 pg/mL PSP: 29.091 ± 18.677 pg/mL Control: 28.444 ± 25.877 pg/mL	ELISA	Saliva
	Oligomeric α-Syn	Oligomeric α-Syn significantly increased in PD patients compared to controls: PD: 0.893 ± 1.949 ng/mL Control: 0.217 ± 0.191 ng/mL	ELISA	Saliva
	Oligomeric α-Syn/ total α-Syn ratio	Oligomeric α-Syn/total α-Syn ratio significantly increased in PD patients compared to controls: PD: 0.235 ± 0.793 Control: 0.0126 ± 0.0079	ELISA	Saliva
12. Shaheen et al., 2020 [40]	Total α-Syn	Total α-Syn levels significantly reduced in PD patients compared to controls: PD: 159.4 ± 61.6 ng/mL Control: 229.9 ± 64 ng/mL	ELISA	Saliva
	Oligomeric α-Syn	Oligomeric α-Syn levels significantly increased in PD patients compared to controls: PD: 47.8 ± 11.8 ng/mL Control: 39.2 ± 9.2 ng/mL	ELISA	Saliva
	Oligomeric α-Syn/ total α-Syn ratio	Oligomeric α-Syn/total α-Syn ratio significantly increased in PD patients compared to controls: PD: 0.35 ± 0.18 Control: 0.19 ± 0.08	ELISA	Saliva
13. Chahine et al., 2020 [41]	Total α-Syn	Total α-Syn non-significantly altered in saliva of PD patients compared to controls: PD: 65.6 ± 42.1 pg/mL Early PD: 49.2 ± 25.4 pg/mL Moderate PD: 63.1 ± 30.3 pg/mL Advanced PD: 83.7 ± 57.9 pg/mL Control: 64.4 ± 60.7 pg/mL	ELISA	Blood CSF Saliva
14. Fernández-Espejo et al., 2021 [42]	Total α-Syn	Total α-Syn levels non-significantly decreased in PD patients compared to controls: PD: 361.89 ± 89 pg/mL Control: 372.1 ± 92 pg/mL	ELISA	Serum Saliva human submandibular gland tissue

ECL: electrochemiluminescence; IP: immunoprecipitation; NS: not specified. Biomarker concentrations of main population groups are underlined.

**Table 5 biomedicines-10-01702-t005:** Quality assessment of case–control studies for amyloid-β and tau according to the customized Newcastle–Ottawa Scale (NOS).

Criteria		1 [18]	2 [19]	3 [20]	4 [21]	5 [22]	6 [23]	7 [24]	8 [25]	9 [26]	10 [27]
Selection	Representativeness of the sample	*	*	-	-	-	*	-	*	*	-
	Sample size	-	-	-	-	-	-	-	-	-	-
	Non-respondents	-	-	-	-	-	-	-	*	-	-
	Ascertainment of the exposure (risk factor)	*	*	*	*	-	-	*	*	*	*
Comparability	Subjects in different outcome (concentration in saliva) groups are comparable (confounding factors controlled)	(*) (-)	(*) (-)	(*) (-)	(-) (-)	(*) (-)	(*) (-)	(*) (-)	(*) (*)	(*) (-)	(-) (-)
Outcome	Assessment of the outcome	*	*	*	*	*	*	*	*	*	*
	Statistical test	*	*	-	*	*	*	*	*	*	*
	Total Score	5	5	3	3	3	4	4	7	5	3
	Quality	fair	fair	poor	poor	poor	fair	fair	good	fair	poor
	Risk of Bias	mod	mod	high	high	high	mod	mod	low	mod	high

* Criteria fulfilled, - Criteria not fulfilled.

**Table 6 biomedicines-10-01702-t006:** Quality assessment of case–control studies for α-synuclein and DJ-1 according to the customized Newcastle–Ottawa Scale (NOS).

Criteria		1 [28]	2 [29]	3 [30]	4 [31]	5 [32]	6 [33]	7 [34]	8 [35]	9 [36]	10 [37,38]	11 [39]	12 [40]	13 [41]	14 [42]
**Selection**	Representativeness of the sample	*	*	-	*	*	*	*	*	*	*	*	*	*	*
	Sample size	-	-	-	-	-	-	-	-	-	-	-	-	*	-
	Non-respondents	-	-	-	-	-	-	-	-	-	-	-	-	-	-
	Ascertainment of the exposure (risk factor)	*	*	-	*	*	*	*	*	*	*	*	*	*	*
**Comparability**	Subjects in different outcome groups are comparable (confounding factors controlled)	(*) (*)	(*) (-)	(*) (*)	(*) (-)	(*) (-)	(*) (*)	(*) (-)	(*) (*)	(*) (*)	(*) (*)	(*) (-)	(*) (-)	(*) (*)	(*) (*)
**Outcome**	Assessment of the outcome	*	*	-	*	*	*	*	*	*	*	*	*	*	*
	Statistical test	*	*	*	*	*	*	*	*	*	*	*	*	*	*
	Total Score	6	5	3	5	5	6	5	6	6	6	5	5	7	6
	Quality	fair	fair	poor	fair	fair	fair	fair	fair	fair	fair	fair	fair	good	fair
	Risk of Bias	mod	mod	high	mod	mod	mod	mod	mod	mod	mod	mod	mod	low	mod

* Criteria fulfilled, - Criteria not fulfilled.

**Table 7 biomedicines-10-01702-t007:** Potential salivary biomarkers associated with AD and PD described in studies.

Biomarker	Morbus Alzheimer	Morbus Parkinson
Aβ42	elevated	
t-tau	unaltered	
p-tau	moderately elevated	
t-tau/p-tau ratio	moderately elevated	
Total α-synuclein		reduced
Oligomeric α-synuclein		elevated
Oligomeric α-Syn/total α-Syn ratio		elevated
DJ-1		unaltered

## Data Availability

Not applicable.

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
