# Peer review of "Systematic Review on Saliva Biomarkers in Patients Diagnosed with Morbus Alzheimer and Morbus Parkinson"

_biomedicines, 2022, doi:10.3390/biomedicines10071702_

Round 1
Reviewer 1 Report
The manuscript is interesting, well balanced and structured. The topic is of wide interest. I just have a minor comment to improve the quality of the manuscript.
I suggest author to re-thinking table styles. In particular tables 1-2 in details, population demographic and biomarkers columns. The most important data are stucked in a small column difficult to read.
Probably invert the colum order from right to the left could be useful.
Also a schematic representation with a summary points before conclusion could be useful to improve the immediately comprehension of the taking home message of this manuscrip
t.
Author Response
Reviewer #1: The manuscript is interesting, well balanced and structured. The topic is of wide interest. I just have a minor comment to improve the quality of the manuscript.
- I suggest author to re-thinking table styles. In particular tables 1-2 in details, population demographic and biomarkers columns. The most important data are stucked in a small column difficult to read. Probably invert the column order from right to the left could be useful.
Thank you very much for this note, both tables were modified in accordance with your suggestion (pages 8-10, between the lines 186-187).
- Also a schematic representation with a summary points before conclusion could be useful to improve the immediately comprehension of the taking home message of this manuscript.
This is, indeed, a relevant addition. Thank you for this idea. A corresponding table with key properties of potential salivary biomarkers has been added (page 26, between lines 557-558).
Reviewer 2 Report
Review of the manuscript entitled: Systematic review on saliva biomarkers in patients diagnosed with Morbus Alzheimer and Morbus Parkinson. The manuscript is interesting but some corrections will be needed. Should be added to the abstract clear purpose of the paper e.g. “The aim of this study is to evaluate/or/study …”
References are needed in lines 61-63,
I believe that a fairly important topic is missing. Nowadays, it is well known that elastin derived peptides (EDPs) which forms during aging induce AD and initiate amyloid deposition. It is believed to be the primary and natural cause of AD. Why did the authors omit this topic? Please complete the manuscript with the latest literature. EDPs can also be detected in body fluids.
Methodology:
I believe that the searched results should only be in English. Why do the authors favor the Russian language? and not, for example, French or Polish? It is well known that the Russians are scientific frauds and the falsification of the results is common there. In the 60-90s of the twentieth century, the USSR reprinted scientific results from the west as its own achievements. Russians do not respect human rights. Therefore, all clinical trials! should be REJECTED. There are no moral and ethical standards in Russia. What is visible every year during sports competitions. The whole country, including scientists, is involved in cheating. Which has been confirmed many times.
I believe the results are well described.
Discussion: The discussion is very well prepared. I think information about EDPs and its correlation with AD should be added.
Conclusions. I believe that there is no need to remind the abbreviations once again, line 636.
Author Response
Reviewer #2: Review of the manuscript entitled: Systematic review on saliva biomarkers in patients diagnosed with Morbus Alzheimer and Morbus Parkinson. The manuscript is interesting, but some corrections will be needed.
- Should be added to the abstract clear purpose of the paper e.g. “The aim of this study is to evaluate/or/study …”
Thank you very much for this note, the statement was modified in accordance with your suggestion (page 1, lines 18-20).
- References are needed in lines 61-63
References are included as suggested by the reviewer.
- I believe that a fairly important topic is missing. Nowadays, it is well known that elastin derived peptides (EDPs) which forms during aging induce AD and initiate amyloid deposition. It is believed to be the primary and natural cause of AD. Why did the authors omit this topic? Please complete the manuscript with the latest literature. EDPs can also be detected in body fluids.
Thank you very much for this note. We assume that there are potential other salivary biomarkers, which might be suitable for diagnosing or monitoring AD and PD. As mentioned by the reviewer, for AD this might include elastin-like or elastin-derived peptides (ELPs or EDPs) of the brain extracellular matrix. These species are found in body fluids and have been discussed to initiate or facilitate AD progression (Ma C et al., Angew Chem Int Ed Engl 2019; Ma J et al., Chemistry 2020; Szychowski KA et al., Cell Mol Neurobiol 2021; Pluta R et al., Antioxidants, 2022). However, we were unable to find studies testing ELPs/EDPs as salivary biomarkers for AD. Therefore, this important topic could not be an essential part of the manuscript. Inspired by the reviewer, we included ELPs/EDPs in the discussion and outlook section of our manuscript (see lines 625-628).
- I believe that the searched results should only be in English. Why do the authors favor the Russian language? and not, for example, French or Polish? It is well known that the Russians are scientific frauds, and the falsification of the results is common there. In the 60-90s of the twentieth century, the USSR reprinted scientific results from the west as its own achievements. Russians do not respect human rights. Therefore, all clinical trials! should be REJECTED. There are no moral and ethical standards in Russia. What is visible every year during sports competitions. The whole country, including scientists, is involved in cheating. Which has been confirmed many times.
The Reviewer is right in many respects. The extent and effects of language bias may have diminished recently because of the shift towards publication of studies in English (this way it is done in Poland, France, or other west oriented countries, where at least the abstract is in English). According to the Cochrane Handbook for Systematic Reviews of Interventions (Chapter 4, section 4.4.5 “Language, date and document format restrictions”), review authors may want, however, to search without language restrictions and decisions about including reports from languages other than English may need to be taken on a case-by-case basis. The authors of the present review have also done it this way, using the advantages of authors' collaboration consisting, inter alia, of multilingual Ukrainians and Germans. Thus, abstracts only available in Russian were sought via eLIBRARY, serving as a database for the majority of articles in Russian (by the way not only from Russia, but also from other CIS countries or former USSR, such as Ukraine, Kazakhstan, Armenia or even Georgia and still from the Baltic States). In the same way, abstracts in Chinese were also considered, two of which were included, in contrast to the studies in Russian, where no relevant studies were found and of course, also not included. The corresponding statement was modified in accordance with your suggestion (page 3, lines 124-130).
- I believe the results are well described. Discussion: The discussion is very well prepared. I think information about EDPs and its correlation with AD should be added.
We mentioned ELPs/EDPs in the discussion (see lines 625-628/, and see also our answer to comment 3).
- I believe that there is no need to remind the abbreviations once again, line 636.
We changed the sentence accordingly (line 641).
Round 2
Reviewer 2 Report
The authors responded properly to my remarks.